# Endoscopic Diagnosis and Management of Barrett’s Esophagus with Low-Grade Dysplasia

**DOI:** 10.3390/diagnostics12051295

**Published:** 2022-05-23

**Authors:** Francesco Maione, Alessia Chini, Rosa Maione, Michele Manigrasso, Alessandra Marello, Gianluca Cassese, Nicola Gennarelli, Marco Milone, Giovanni Domenico De Palma

**Affiliations:** 1Department of Clinical Medicine and Surgery, University of Naples “Federico II”, 80131 Naples, Italy; alessiachini@hotmail.it (A.C.); alessandramarello@gmail.com (A.M.); gianluca.cassese91@gmail.com (G.C.); nicogenna@yahoo.it (N.G.); milone.marco.md@gmail.com (M.M.); giovanni.depalma@unina.it (G.D.D.P.); 2Department of Advanced Biomedical Sciences, University of Naples “Federico II”, 80131 Naples, Italy; michele.manigrasso89@gmail.com

**Keywords:** Barrett’s esophagus, low-grade dysplasia, endoscopic mucosal resection, radiofrequency ablation, cryotherapy, endoscopy

## Abstract

Barrett’s Esophagus is a common condition associated with chronic gastroesophageal reflux disease. It is well known that it has an association with a higher incidence of esophageal adenocarcinoma, but this neoplastic transformation is first preceded by the onset of low and high-grade dysplasia. The evaluation of low grade dysplastic esophageal mucosa is still controversial; although endoscopic surveillance is preferred, several minimally invasive endoscopic therapeutic approaches are available. Endoscopic mucosal resection and radiofrequency ablation are the most used endoscopic treatments for the eradication of low-grade dysplasia, respectively, for nodular and flat dysplasia. Novel endoscopic treatments are cryotherapy ablation and argon plasma coagulation, that have good rates of eradication with less complications and post-procedural pain.

## 1. Introduction

Barrett’s Esophagus (BE) is a condition defined as a metaplastic transformation of the distal normal esophageal squamous mucosa to intestinal columnar epithelium [1]. The most important risk factor for this condition is non-treated and long-term gastroesophageal reflux disease (GERD), developing in approximately 5% to 15% of patients that undergo endoscopic examination with a clinical diagnosis of GERD [2]. BE is recognized as the only established condition that increases the risk of esophageal adenocarcinoma (EAC) [3]. Risk factors associated with increased incidence of BE are male sex, weight gain, age >50 years old, hiatal hernia. The neoplastic progression from dysplasia to carcinoma consists of a multistep process that takes a variable time to develop, during which mucosal alterations occur [4]. For patients with high grade dysplasia (HGD), all the international guidelines suggest the necessity of the endoscopic eradication of dysplasia because of the high rate of progression to adenocarcinoma. On the other hand, the best management for patients with low grade dysplasia (LGD) in Barrett is still controversial, first because is an over-diagnosed condition, and second because the risk of progression to HGD and EAC is variable [3,5].

## 2. Diagnosis

The gold standard for BE diagnosis is endoscopy. The detection of metaplastic columnar epithelium at least 1 cm in the distal esophageal mucosa can be achieved with many optical methods, although certain diagnosis is performed by the histological examination of mucosal biopsies. BE appears on esophagogastroduodenoscopy as a salmon-coloured mucosa extending more than 1 cm proximal to gastroesophageal junction [3]. During endoscopy, according to Praga classification, circumferential extension (C value) of BE and its maximum extension (M value) should be described from the gastroesophageal junction; the metaplastic mucosa may endoscopically present a segmental or circumferential appearance [6] (Figure 1).

When endoscopist suspects a diagnosis of BE, biopsies following Seattle Protocol must be performed that involve 4-quadrant biopsy sampling every 2 cm throughout the columnar-lined esophagus. Furthermore, every mucosal irregularity should be sampled because in these areas it is more probable to find dysplastic tissue [7].

At histological examination, cells with BE-associated LGD appear with mild architectural abnormalities, increased numbers of mitotes and cytologic atypia, such as increased nuclear: cytoplasmic (N:C) ratio and nuclear elongation [8].

Both in USA and Europe, the most used grading system for the histopathological diagnosis of LGD is the modified Vienna classification, which reports five classes of epithelial changes in gastrointestinal dysplasia [9] (Table 1).

However, there is no accordance between pathologists regarding the exact histological definition of LGD, because this grading system is based on few categories that do not allow to distinguish the various types of LGD.

A recent study reported that in clinical practice not all LGDs are the same, and only a small percentage of these progress to HGD/EAC. This condition may depend on the similar histological features between low-grade dysplastic and inflammatory tissue, which make histological diagnosis non-specific [10]. For this reason, some histological criteria, such as an increase of mitosis, mucin depletion, a nuclear enlargement and the loss of surface maturation of cells, allow better prediction than others on the progression risk to HGD/EAC [11].

In particular, there are some differences in the interpretation of the histological findings between American and European pathologists. Vennalaganti et al. [12], in their multicenter prospective study, confirmed this discordance, showing that European pathologists tended to diagnose fewer cases of LGD with predominant inflammatory features (LGD-I) using fewer criteria for the definition of dysplasia compared with American pathologists, and this could explain the increased rate of progression to HGD/EAC in the European studies.

Concerning LGD with predominant inflammatory features, the Expert Review from the Clinical Practice Updates Committee of the American Gastroenterological Association (AGA, Bethesda, MD, USA) recommended not performing surveillance biopsies in patients with active esophageal inflammation and suggested repetition of endoscopic examinations after an anti-reflux regimen to avoid an overestimation of LGD [13].

A high-quality endoscopic exam, although it does not replace the histological diagnosis, can increase dysplastic lesions detection rate [14]. Advanced imaging technologies can show an irregular appearance of the esophageal mucosal surface with a high accuracy. Furthermore, random biopsies performed with white light endoscopy may not be completely accurate in the recognition of all dysplastic areas, that often present a focal distribution, sampling only 4–5% of esophageal mucosa [15].

New, advanced imaging technologies can lead to targeted biopsies being developed to improve early diagnosis of dysplastic Barrett in surveillance endoscopic programs, and targeting the biopsy mapping of areas of greatest concern. Among these innovative techniques, the most used are chromoendoscopy, virtual chromoendoscopy, confocal endomicroscopy and artificial intelligence.

Conventional chromoendoscopy consists of topical application of dyes through a spray catheter to enhance atypical features of dysplastic mucosa more than in white light endoscopy [16]. Methylene Blue (MB) is the most used dye for BE detection, because it is selectively absorbed into intestinal type cells. Canto et al. [17] showed that the accuracy of methylene blue to detect specialized columnar epithelium was 95%, using histology as a reference standard. Acetic acid is a weak acid, also used as non-absorbed contrast agent for the upper gastrointestinal tract, which induces an acetowhitening reaction on the esophageal mucosa. In dysplastic BE, this whitening reaction is followed by a focal redness, defined by Longcroft-Wheaton et al., as a predictor of pre-neoplastic lesion [18].

In their prospective study, Longcroft-Wheaton et al. [19] showed that, in 132 patients, neoplastic and dysplastic mucosal areas lost the acetowhitening reaction faster than non-neoplastic mucosa, and that this is an objective method to evaluate suspected mucosal area.

Recently, the evolution of digital endoscopy allowed the development of the virtual chromoendoscopy, an advanced imaging technique that allows the endoscopist to enhance the visualization of the BE using specific wavelengths of light. Narrow band imaging (NBI) uses green and blue lights to improve superficial mucosal details, such as vascular patterns and mucosal irregularities, without the use of contrast agents (Figure 2). Competing technologies are i-SCAN and FUJI intelligent chromo endoscopy, which use digital filters for image acquisition with white light, enhancing vascular mucosal patterns [16].

Several studies reported the advantage of virtual chromoendoscopy in comparison to traditional chromoendoscopy, both for the lower costs and for the shorter time procedure related to the conventional technique. Furthermore, virtual chromoendoscopy has the advantage to enhance vascular patterns, unlike the conventional chromoendoscopy [20].

In their meta-analysis, Mannath et al. [21] reported that a Narrow Band Imaging with a sensitivity greater than 90% proved to be an effective screening method and an accurate technique to perform targeted biopsies.

Quemseya et al. [22] showed that advanced imaging technologies increased the diagnostic yield by 34% compared to white light endoscopy, without a significant difference between chromoendoscopy and virtual chromoendoscopy.

Confocal endomicroscopy (CLE) is another advanced technology that allows the detection of BE using a low-power light laser. The final image is derived from the reflection of this light from the pointed surface. There are two systems of CLE: endoscopy (eCLE; Pentax, Toshima, Japan), now commercially available, and a probe-based system (pCLE), that can be used with a standard endoscope [23]. The high-resolution microscopic visualization of mucosa is achieved by the intravenous infusion of a fluorescent contrast agent (usually fluorescein). This visualization method has a high rate of LGD prediction, decreasing the number of biopsies required for the histological examination [24].

At CLE examination, the crypt architecture of BE is characterized by intermittent dark mucin in goblet cells in the upper parts of the mucosal layer. In the deeper parts, villous, dark, regular cylindrical Barrett’s epithelial cells are present [25].

As reported in several studies, pCLE has proven to be an effective diagnostic technique in dysplastic and neoplastic detection, with a sensitivity of 62.5% compared to the 32.7% related to wight light endoscopy. Furthermore, pCLE has a high negative predictive value (NPV) regarding BE and related dysplasia and neoplasia diagnosis, and a non-inferior negative predictive value compared to 4-quadrant biopsies screening [26,27,28]. Dunbar et al. [29] in their prospective trial demonstrated an increased detection rate of neoplasia for pCLE compared with the standard biopsy protocol.

Kiesslich et al. [30] created a confocal classification system that subdivides three types of cells at the distal esophagus: gastric-type epithelium, Barrett’s epithelium, and neoplastic, based on the microscopic mucosal architecture of cells and vessels in patients with dysplastic BE. According with these criteria, the sensitivity for the detection of dysplasia in these patients was 93%.

Despite the great enthusiasm for this advanced technology, in clinical practice pCLE is not much used because of long training times higher costs [16].

The American Society of Gastrointestinal Endoscopy (ASGE) conducted a large metanalysis to evaluate costs and benefits of advanced imaging technologies for the detection of BE. The results supported the use of traditional chromoendoscopy, NBI and pCLE to perform targeted biopsies in endoscopic surveillance, decreasing the number of biopsies and improving accuracy with a lower procedure time [31].

New perspectives for advanced diagnostics of BE opened up with the advent of artificial intelligence, that uses colour and texture filters to detect pre-neoplastic and neoplastic lesions by comparing with a computer algorithm. Recent data showed that artificial intelligence had a sensitivity of 90% and a specificity of 88% for the detection of LGD [32].

## 3. Management

It’s difficult to understand the natural progression of the histological diagnosis of LGD and its neoplastic and clinical outcomes. Weston et al, conducted a study showing that the progression rate of BE from LGD to HGD or EAC was 6%. In this trial, the diagnosis of LGD was verified by a single pathologist [33]. However, this progression rate may not be reliable, because LGD, initially diagnosed by a single pathologist, was not always confirmed on endoscopic follow-up. Conio et al. [34] showed that among patients with histological diagnosis of LGD at the first endoscopic exam, dysplasia was not confirmed in 75% of cases on subsequent histological controls. Failure to confirm LGD during surveillance can be explained by there being similar cytologic features between inflammation-mediated injury and dysplasia, which makes it difficult to differentiate these two conditions and leads to an overdiagnosis of LGD [12]. Therefore, various studies have affirmed that the agreement between two or more experienced pathologists for the diagnosis of LGD increased the risk of progression to HGD and EAC [35,36].

It is critically important to achieve a reproducible diagnosis of LGD for its impact in the dysplastic progression and, consequentially, for its management. Actually, a confirmed diagnosis of LGD involves a significant risk of neoplastic incidence, with a progression rate to HGD and EAC from 0.5% to 13% per year [36].

Currently, there is not a consensual approved management for the Barrett’s LGD. The American College of Gastroenterology (ACG) Guidelines recommended endoscopic therapy for patients with confirmed LGD, even if endoscopic surveillance is still indicated [3]. The European Society of Gastrointestinal (ESGE, Leuven, Belgium) Guidelines suggest that endoscopic eradication of LGD should be proposed to patients with a histological diagnosis confirmed by a pathologist with special interest in gastrointestinal pathology, after a surveillance interval of six months [37]. Similarly, ASGE and American Gastroenterological Association (AGA) in their latest guidelines consider the endoscopic eradication of LGD a reasonable option in patients with a diagnosis confirmed at a second examination within 3 to 6 months, even if both eradication therapy and surveillance are indicated [38,39].

In Japan, the histopathologic evaluation of LGD differs from in Western Countries, because this grade of dysplasia is defined as *well-differentiated adenocarcinoma with low-grade*
*atypia (non-invasive)*. In the current status, Japan Gastroenterological Endoscopy Society (JGES) guidelines recommend performing endoscopic resection for HGD, while a 6-monthly endoscopic surveillance for LGD conditions is preferred [40,41].

Several studies compared the rate of LGD progression between patients undergoing endoscopic treatment and endoscopic surveillance. Klair et al. [42], in their large meta-analysis, included 543 patients with LGD and evaluated the progression risk of dysplasia between patients treated with RFA and those under surveillance. In the RFA group, 11 patients developed HGD or EAC, while 71 patients of surveillance group progressed to HGD or EAC, showing a significantly lower rate of progression in patients undergoing treatment.

Phoa et al. [43] published a randomized clinical trial showing that the risk of progression to HGD in patients undergoing an ablative treatment was less likely than the control group (1.5% ablation group (*n* = 1) vs. 26.5% control group (*n* = 18), *p* < 0.001). Recent studies by Small et al. [44] and Barret et al. [45] supported this data, showing that a timely treatment from the first diagnosis of LGD is associated with a lower rate of progression of dysplasia and a high percentage of complete eradication of dysplasia (CE-D), that is defined as no further detection of dysplasia on endoscopic biopsies subsequent the initial diagnosis (Table 2).

The first line treatment of dysplastic Barrett should be endoscopic eradication therapy (EET) [49]. Various endoscopic techniques can be used for the endoscopic eradication for dysplastic BE, including resective treatments such as Endoscopic Mucosal Resection (EMR) and Endoscopic Submucosal Dissection (ESD), and ablative treatments such as Radiofrequency ablation (RFA), cryotherapy ablation and Argon Plasma Coagulation (APC), or a combination of these techniques. The endoscopic therapeutic approach depends on the appearance of the esophageal mucosa affected by dysplasia. For non-nodular Barrett Dysplasia, ablative treatment is the gold standard, and the most preferred ablative technique is RFA [50]. Patients with nodularity BE should undergo EMR of the lesion. EET can be achieved with ablative endoscopic therapy of the remaining BE [8].

**EMR and ESD**. EMR and ESD are resective techniques used for the treatment of nodular lesions of the esophageal mucosa. These procedures are considered an effective and less invasive alternative to esophagectomy, which is associated with a higher risk of mortality and morbidity [51].

EMR is a procedure based on the endoscopic snare resection of the flat mucosal lesion preceded by injection of a saline solution into the submucosal layer to lift the lesion area away from the deeper muscular layer [52].

EMR efficacy in patients with LGD has been shown in many clinical trials. Konda et al. [53] demonstrated a rate of 95.9% for complete remission of dysplasia, with a recurrence of 8.1% after 33 months of follow-up. Similar results have been reported by Gerke et al. [54], who paid more attention to complication rate related to EMR. In their retrospective single-centre study, among 41 patients treated with EMR for BE eradication, 65% of patients had a complication after the procedure, and, in particular, 2 patients had esophageal perforation, while, in 18 patients, esophageal strictures occurred. The most common complications of endoscopic resection performed for BE treatment is stricture, followed by bleeding and perforation. The incidence of strictures seems to increase with the length of BE segment and when a single-session circumferential resection of more than >75% of the circumference is performed [55].

ESD is an advanced technique that can be used in selected cases, requiring a higher level of experience compared to EMR. Furthermore, ESD has a higher complication rate, which results unnecessarily in most cases of LGD [56].

The advantage of endoscopic resection over ablation techniques is to allow a histological examination of the resected specimen, defining the grade of dysplasia and the appropriate treatment [57]. Several studies reported a relevant change of histopathologic staging of dysplastic Barrett’s mucosa from 25% to 35% in patients undergoing EMR, supporting this endoscopic treatment for a more accurate therapeutic management [58,59]. This technique could be recommended for any visible abnormality of esophageal mucosa, but data in the literature on the efficacy of EMR for patients with flat dysplasia are limited.

Resective endoscopic techniques are non-destructive procedures with the advantage of accurate histopathological analysis of the dysplastic resected tissue. However, these procedures are characterized by a longer operative time and a higher complication rate compared to ablative endoscopic treatment.

**RADIOFREQUENCY ABLATION.** Radiofrequency ablation (RFA) is the most performed ablative treatment for BE with LGD. RFA is an endoscopic technique that uses thermal energy to induce necrosis of esophageal dysplastic mucosa and regeneration of normal mucosa [50]. This system consists of an energy generator, which is a bipolar electrode array with 60 tightly spaced electrodes encircling the balloon catheters. RFA can be performed with two different ablation devices, for focal or circumferential ablation. The circumferential balloon (HALO 360) is indicated for long segment and circumferential BE, while the focal balloon (HALO 90) works better for focal lesions with a difficult anatomical localization, as with a hiatal hernia [1]. Despite the choice of the focal or circumferential device according to the morphology of dysplastic lesion, a comparative study between these showed that focal balloon seems to be more effective compared to the circumferential balloon, allowing a significant reduction of the lesion at the first treatment session with a fewer number of sessions to achieve LDG eradication [60].

The first study that evaluated RFA efficacy was the AIM dysplasia trial [47], which included patients with non-nodular dysplastic BE who were randomized to RFA or a sham procedure. This multicentric trial showed that after 12 months of follow up, 90.5% of patients with LGD treated with RFA achieved CE-D compared to 22.7% of the control group. Orman et al. [61] compared 14 studies evaluating the efficacy and durability of RFA for treatment of LGD; the eradication of dysplasia was achieved in 91% of patients.

Other studies demonstrated the efficacy of RFA to eradicate LGD, showing that patients treated with RFA eradicated CE-D in 91–95% of cases [43,44,62,63].

RFA is indicated for the ablation of residual or metachronous dysplastic BE even in association with endoscopic resection techniques [64]. In a prospective cohort trial, 23 patients underwent 25 endoscopic resections before RFA. In 20 of 21 patients with residual low-grade intraepithelial neoplasia/high-grade intraepithelial neoplasia (LGIN/HGIN) after endoscopic resection and before ablation, complete remission of neoplasia (CR-neoplasia) was achieved with RFA (95%), proving it to be a safe treatment with a low rate of adverse events [46]. The English RFA registry revealed a significant improvement in the achievement of CE-D when RFA was preceded by EMR in a cohort of 500 patients, with a CR-D in patients with LGD of 86% after 6-years follow-up [65].

The safety profile of RFA is higher compared to other endoscopic therapies. Chadwick et al. [66] compared the safety of RFA and complete EMR in dysplastic BE, showing that RFA and complete EMR were equally effective but EMR was associated with a higher complication rate. The most common adverse event in patients underwent RFA is esophageal stricture, as reported in the metanalysis conducted by Qumseya et al. [67], although less frequently, haemorrhage, chest pain and perforation may occur.

Recurrence after this ablative endoscopic treatment is possible, and patients treated with RFA should be closely followed up [68]. The literature reports a variable rate of recurrent or persistent intestinal metaplasia after RFA. Gupta et al. [69] observed an incidence of recurrence after 1 year of 20%, and after 2 years of 33%. A lower rate of recurrence was showed by Orman et al. [70], with detection of intestinal metaplasia at follow-up of 7%. Recent studies confirmed that the annual rate of recurrence of BE after RFA was 5–8%, with a dysplasia recurrence of 0.9–2%/year [71,72,73].

**CRYOTHERAPY ABLATION.** Although RFA remains the most frequently used endoscopic ablative treatment for dysplastic BE, cryotherapy ablation is an emerging technique that consists of cycles of rapid cooling with a cryogen, such as liquid nitrogen or carbon dioxide, (unlike RFA that uses heat energy), that induces tissue necrosis of esophageal dysplastic mucosa [74,75,76].

This is a non-contact technique, more targeted than RFA, and can explain the lower risk of stricture incidence compared to RFA [77]. Moreover, cryotherapy can ablate deeper to the submucosa, while RFA is limited to muscularis mucosa [78]. There are three systems for the cryotherapy ablation technique: the first uses carbon dioxide as cryogen through a spray catheter; the second, that is the most used, is based on liquid nitrogen through a spray catheter; the last and more recent system employs nitrous oxygen to ablate esophageal mucosa through a balloon device [50].

In a study conducted by Ghorbani et al. [79], 96 patients with LGD and HGD were treated with the CryoSpray Ablation System using liquid nitrogen. Among patients with LGD, 91% completely achieved the eradication of dysplastic mucosa. Mohan et al. [80] clearly displayed the outcomes of cryoablation treatment using liquid nitrogen, showing a CE-D rate of 81–85%. Despite the absence of level 1 evidence supporting cryotherapy treatment of BE, various studies showed cryoablation efficacy as first line therapy for LGD. Hamade et al. [81] in their metanalysis collected data from six studies revealing that 97.9% of the patients achieved complete eradication of dysplasia; similar rates were reported by Tariq et al. [82].

The efficacy and safety of the most recent cryoablation technique, the Cryoballoon Ablation, was investigated in metanalysis conducted by Westervald et al. [83], which demonstrated the achievement of complete eradication of metaplasia (CE-M) and CE-D in 88% and 94% of patients with LGD, respectively.

Table 3 reports several studies showing CE-D rates between resective and ablative treatments. Notably, it shows a similar efficacy between RFA and cryotherapy, reporting a comparable rate of CE-D as initial therapy (Table 3).

In this regard, a retrospective cohort study conducted by Fasullo et al. [84], evaluated the clinical outcomes of RFA, compared to the most used liquid nitrogen spray cryotherapy (LNSC), in patients with dysplastic BE, and a similar rate of CE-D between RFA and LNSC groups was reported (81% vs. 71%). The LNSC group required a higher number of sessions compared to RFA group to achieve CE-D. Similar results were reported by Thota et al. [85], supporting the use of cryotherapy, especially in older patients with comorbidities.

Solomon et al. [86] focused attention on the incidence of postprocedural pain between RFA and cryotherapy ablations. The numeric pain scale score was evaluated between the two ablation modalities groups, showing immediate and 48-h-post procedural pain scores being lower in patients treated with Cryoablation. In the literature, data on Focal Cryoballon ablation (CRYO) of dysplastic BE are still limited. Van Munster et al. [87] showed a similar efficacy between a single session of RFA and CRYO, but patients treated with CRYO reported less post-procedural pain compared with RFA. Often, the length of Barrett’s dysplastic mucosa and the hiatal hernia interferes with the achievement of complete eradication in patients treated with RFA; therefore, there may be a role for cryotherapy in patients who fail to respond to initial RFA [88,89]. A single center retrospective study demonstrated the efficacy of the salvage cryotherapy with liquid nitrogen in patients with recurrence of LGD after RFA, with a CE-D of 83% (5/6 patients) [88]. Trinidade et al.’s study [90] supports Sengupta’s data.

**Argon Plasma Coagulation.** Argon plasma coagulation (APC) is an ablation technique based on the induction of dysplastic tissue necrosis using a probe-based catheter with jet-ionized argon gas. Although it has been used in the gastrointestinal tract and for BE treatment for his effectiveness, this technique in the esophageal mucosa has a high risk of perforation. For this reason, recently an advanced form of this ablative technique has been developed, i.e., Hybrid-APC, which combines APC with a preventive saline submucosal injection [91].

The safety and efficacy of Hybrid-APC have been confirmed by several studies, which mainly included cases of refractory BE. Manner et al. [91] conducted a study including 60 patients with residual BE-related dysplasia after endoscopic resection; 96% of these achieved complete macroscopic remission after Hybrid-APC treatment, with a low rate of stricture formation, which represents the most frequent adverse event. Estifan et al. [92] showed a particular indication of Hybrid-APC: an 81-years old patient underwent ESD for an esophageal nodule with HGD, was treated with Hybrid-APC for the residual low dysplastic lateral margins of lesion, reaching the complete eradication of residual intestinal metaplasia and associated dysplasia. Recently, Wronska et al. [93] conducted a large study for the evaluation of short and long-term outcomes of Hybrid-APC in LGD. They showed a complete ablation rate of 60–78% at 6 weeks, and a CE-D rate of 93% after 2 years-follow up in patients treated with one or more APC sessions.

**Table 3 diagnostics-12-01295-t003:** Characteristics of studies comparing the different techniques for LGD BE-related treatment.

	Study	Patients	Achievement of CE-D (%)	Recurrence after CE-D (%)	Number of Sessions for CE-D (Median)
Konda et al. [53]	EMR	107	80% (80/107)	8.1% (9/107)	**-**
Gerke et al. [54]	EMR	41	85.4% (36/41)	21.9% (9/41)	2.4
Shaheen et al. [47]	RFA	42	90.5% (38/42)	5% (2/42)	-
Shaheen et al. [62]	RFA	52	98% (51/52)	-	-
Phoa et al. [43]	RFA	68	92.6% (63/68)	1.5% (1/68)	-
Small et al. [44]	RFA	45	95.6% (43/45)	2.2% (1/45)	2
Vliebergh et al. [63]	RFA	342	93% (318/342)	-	2
Ghorbani et al. [79]	LNSC	23	91% (21/23)	-	2.9
Westerlvald et al. [83]	CRYO	75	93.8% (70/75)	-	1.5
Thota et al. [85]	RFA	73	87.5% (63/73)	11.1% (7/73)	3
	LNSC	81	78.9% (63/81)	14.3% (9/81)	3
Fasullo et al. [84]	RFA	100	81% (81/100)	11.1% (11/100)	2.5
	LNSC	62	71% (44/62)	13.6% (8/62)	4.8
van Munster et al. [87]	RFA	26	90% (23/26)	-	-
	CRYO	20	88% (17/20)	-	-
Manner et al. [91]	Hybrid-APC	50	96% (48/50)	-	3.5
Wronska et al. [93]	Hybrid-APC	71	93% (66/71)	4% (3/71)	2

EMR: Endoscopic Mucosal Resection. RFA: Radiofrequency Ablation. LNSC: Liquid Nitrogen Spray Cryotherapy. CRYO: balloon-based focal cryoablation. Hybrid-APC: Hybrid-Argon Plasma Coagulation. CE-D: Complete Eradication of Dysplasia.

## 4. Conclusions

LGD associated with BE is a controversial condition, that does not have a consensual management line. The rate of progression to HGD and EAC is variable, but a certain diagnosis of low-grade dysplasia is a predictive risk of progression to adenocarcinoma. For this reason, patients with LGD diagnosis should be referred to an endoscopic treatment and follow-up. Several options are available for the eradication of the esophageal dysplasia. The ablation methods are usually used for flat dysplastic BE, even if the resection techniques allow a confirmed diagnosis by histological examination. Endoscopic surveillance is recommended in patients not undergoing endoscopic treatment.

## Figures and Tables

**Figure 1 diagnostics-12-01295-f001:**
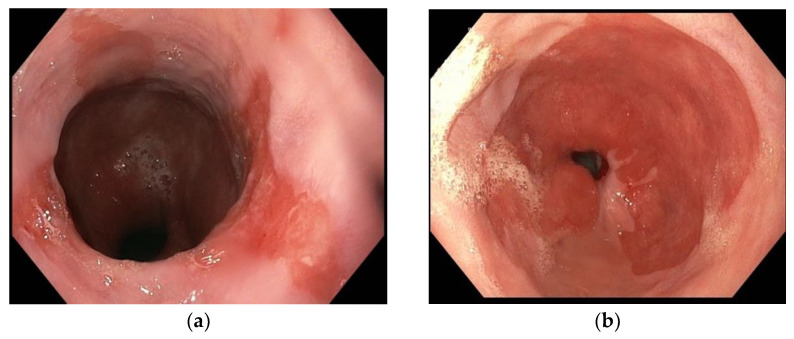
Gastroesophageal junction in Barrett’s Esophagus; (**a**) Segmental Barrett’s Esophagus (**b**) Circumferential Barrett’s esophagus.

**Figure 2 diagnostics-12-01295-f002:**
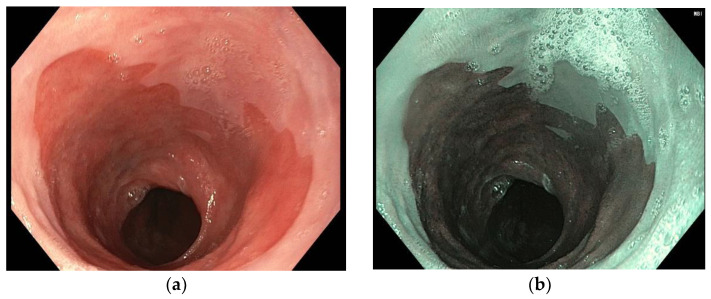
Endoscopic evaluation of circumferential Barrett’s Esophagus (**a**) in white light endoscopy; (**b**) in narrow band imaging vision (NBI).

**Table 1 diagnostics-12-01295-t001:** Revised Vienna Classification.

Category	Diagnosis
**1**	Negative for neoplasia (ND)
**2**	Indefinite for neoplasia (ID)
**3**	Mucosal low-grade neoplasia (LGD) Low-grade adenoma Low-grade dysplasia
**4**	Mucosal high-grade neoplasia (HGD)
4.1	High-grade adenoma/dysplasia
4.2	Non-invasive carcinoma (*carcinoma in situ*)
4.3	Suspicious for invasive carcinoma
4.4	Intramucosal carcinoma
**5**	Submucosal invasion by carcinoma

**Table 2 diagnostics-12-01295-t002:** Progression of LGD to HGD and EAC between patients treated with RFA and Surveillance.

	Study	Patients	Time of Follow-Up	Ratio of Disease Progression with RFA to HGD	Ratio of Disease Progression with RFA to IMC/EAC	Ratio of Disease Progression with Surveillance to HGD	Ratio of Disease Progression with Surveillance to IMC/EAC	CE-D in RFA Group (%)	CE-D in Surveillance Group (%)
Barret et al., [45]	Randomized trial	82	36 months	5/40	-	11/42	-	-	-
Phoa et al., [43]	Randomized trial	136	36 months	1/68	1/68	18/68	6/68	98.4%	27.9%
Pouw et al., [46]	Randomized trial	136	22 months	0/68	0/68	4/68	1/68	95%	-
Shaheen et al., [47]	Randomized trial	64	12 months	2/42	0/42	3/22	0/22	90.5%	22.7%
Small et al., [44]	Retrospective study	170	28 months	0/45	1/45	29/125	7/125	95.6%	31.2%
Kahn et al., [48]	Retrospective study	173	90 months	7/79		14/94		97.5%	61.7%

LGD = Low Grade Dysplasia; HGD = High Grade Dysplasia; RFA = Radiofrequency Ablation; CE-D = Complete Eradication of Dysplasia.

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
