# Peer review of "Endoscopic Diagnosis and Management of Barrett’s Esophagus with Low-Grade Dysplasia"

_diagnostics, 2022, doi:10.3390/diagnostics12051295_

Round 1
Reviewer 1 Report
The article was well reviewed about diagnosis and management of Barrett’s Esophagus (BE). It consisted of two parts both diagnostic part and management part. At diagnostic part, the authors summarized several advanced diagnostic technologies for improving early phase dysplastic BE. At management part, the author explained various endoscopic techniques represented by EMR and ESD, and ablative treatments such as Radiofrequency ablation, Cryotherapy ablation and Argon Plasma Coagulation.
This information seems to be important for the endoscopists.
However, there are some problems in this paper.
Discordance existed among pathologists in USA and Europe in diagnosis of low-grade dysplasia (LGD) for patients with BE. How accurate can we diagnose LGD?
If the author will support comprehensive conditions, you have to quote other large organization’s guidelines, such as American Society of Gastrointestinal Endoscopy (ASGE), American Gastroenterological Association (AGA) and Japan Gastroenterological Endoscopy Society (JGES). For example, AGA’s expert reviewed BE patients with confirmed low-grade dysplasia, a repeat examination with high-definition white-light endoscopy should be performed within 3-6 months to rule out the presence of a visible lesion, which should prompt endoscopic resection. Both BE Therapy (BET) and continued surveillance are reasonable options for the management of BE patients with confirmed and persistent low-grade dysplasia. You can refer the later, Gastrointest Endosc 2016 83(4):684-98, Gastroenterology 2020;158:693–702, and Dig. Endosc 2020; 32: 452-493.
Is it too late to perform ESD/EMR after detection of HGD or EAC progression? Should we eradicate early phase dysplastic BE like LGD? The other organizations recommend short span surveillance than intervention for BE with LGD. Are these interventions worth effort?
Table 1 dealt with one pooled analysis, one RCT trial, and one retrospective study as the same observation period. The longer the observation period, the higher the frequency of occurrence. It is necessary to clarify the frequency of occurrence by using person-year during the observation period.
Table2 is very immature and inconsistent. Which is intervention group RFA or CRYO? Which is comparison group RFA or CRYO? What is “n circle days”? The author must explain the meaning of abbreviations.
There are many spell mistakes and grammatical errors throughout, so it will be better to do English proofreading.
Author Response
Response to Reviewer 1 Comments
Point 1: Discordance existed among pathologists in USA and Europe in diagnosis of low-grade dysplasia (LGD) for patients with BE. How accurate can we diagnose LGD?
Response 1: The diagnosis of LGD, as descripted in the manuscript, is difficult to obtain with a high accordance between pathologists. The similar histological features between LGD and inflammation may be confusing and can lead to a non-unique diagnosis. I’ve provided to add in the manuscript a digression on the discordance existing among pathologists in USA and Europe, including also more details on histological diagnosis of LGD in the Diagnosis section.
Point 2: If the author will support comprehensive conditions, you have to quote other large organization’s guidelines, such as American Society of Gastrointestinal Endoscopy (ASGE), American Gastroenterological Association (AGA) and Japan Gastroenterological Endoscopy Society (JGES). For example, AGA’s expert reviewed BE patients with confirmed low-grade dysplasia, a repeat examination with high-definition white-light endoscopy should be performed within 3-6 months to rule out the presence of a visible lesion, which should prompt endoscopic resection. Both BE Therapy (BET) and continued surveillance are reasonable options for the management of BE patients with confirmed and persistent low-grade dysplasia. You can refer the later, Gastrointest Endosc 2016 83(4):684-98, Gastroenterology 2020;158:693–702, and Dig. Endosc 2020; 32: 452-493
Response 2: I proceeded to quote ASGE guidelines in the diagnosis section supporting the use of advanced imaging technologies. In the management section, I have added ESGE, AGA and JGES guidelines regarding the management of LGD, to better support comprehensive conditions
Point 3: Is it too late to perform ESD/EMR after detection of HGD or EAC progression? Should we eradicate early phase dysplastic BE like LGD? The other organizations recommend short span surveillance than intervention for BE with LGD. Are these interventions worth effort?
Response 3: International Gastrointestinal Societies recommend both surveillance than eradication for LGD, as reported in the manuscript. Despite this is an early dysplastic stage, there is a consistent progression risk to HGD and EAC for confirmed diagnosis of LGD; then, even if surveillance remains an option for LGD, the best management for dysplastic Barrett should be the eradication therapy. Ablative treatments are preferred for LGD treatment, but only endoscopic resection (EMR/ESD) allows a histological sample for a confirmed diagnosis, with a resulting accurate staging that provides to a more accurate management.
Point 4: Table 1 dealt with one pooled analysis, one RCT trial, and one retrospective study as the same observation period. The longer the observation period, the higher the frequency of occurrence. It is necessary to clarify the frequency of occurrence by using person-year during the observation period.
Response 4: I have added a column in Table 1, specifying the time of follow-up of recruited patients.
Point 5: Table 2 is very immature and inconsistent. Which is intervention group RFA or CRYO? Which is comparison group RFA or CRYO? What is “n circle days”? The author must explain the meaning of abbreviations.
Response 5: I’ve better explained table’s definitions and abbreviations, adding explanation of the abbreviations under the table and improving the table’s title. The definition “n° days” means the number of days during which patients had postprocedural pain from the treatment; I have explained this using “n° days from the procedure”. However, I could not add other studies in this table, because there are few studies that directly compare RFA and Cryoablation
Point 6: There are many spell mistakes and grammatical errors throughout, so it will be better to do English proofreading.
Response 6: I have corrected spell mistakes and grammatical errors, doing English proofreading. I’ve also modified many parts of the manuscript to allow a better understanding.
Reviewer 2 Report
Dear Authors
The manuscript is a theme review that does not have quality to be published in its present form. It lacks clarity, is not well structured and approaches several issues in a superficial way, while in others it could be much more concise. Moreover, the text lacks clarity and coherence that eventually may come from extensive editing of English language and style.
Author Response
Response to Reviewer 2 Comment
Point 1: The manuscript is a theme review that does not have quality to be published in its present form. It lacks clarity, is not well structured and approaches several issues in a superficial way, while in others it could be much more concise. Moreover, the text lacks clarity and coherence that eventually may come from extensive editing of English language and style
Response 1: Dear Reviewer,
I have modified many parts of the manuscript to better explain some unclear concepts; I’ve also edited English language and corrected grammatical errors. I hope that your requests have been met; otherwise, I will modify the lacking parts if you will show them to me.
Round 2
Reviewer 1 Report
This paper has been substantially revised after we pointed out some revisions.
Overall, I have an impression that too many abbreviations were used. There was no need to use abbreviations for things that were used infrequently. If it were not so common, it would be illegible for readers.
The risk of going from LGD to HGD/EAC is relatively high; if LGD could be evaluated with high accuracy without including inflammation, it was worthwhile to eradicate LGD early on.
It was clear that some LGD cases probably progress to HGD, which was explained in Table2.
The author misinterpreted the wording of the Japanese guideline, which recommends endoscopic treatment for HGD, not LGD.
My opinion was on whether it was wrong to perform ESD or other aggressive treatments in the cases of LGD.
To be precise and detailed, it would be appropriate to compare CE-D between the LGD observation groups and the prior treatment groups over a long period. An argument should be made regarding this.
As for Table 3, the problem is not that there are not enough papers, but that the table is not complete and may be difficult for readers to read. The comparison and target groups are not consistent, and the expressions were lacks politeness and care. Abbreviations and non-abbreviated expressions are mixed in the table. Could we really say that this table was complete? I know what “N circle” means. But is it common? I said that you had to fix it so that readers can easily understand this. Complication and pain do not need to be included in the table, they should be expressed in the text. Rather, please make a more comprehensive table comparing CE-D for ESD/EMR, RFA, CRYO, and APC.
Author Response
Point 1: Overall, I have an impression that too many abbreviations were used. There was no need to use abbreviations for things that were used infrequently. If it were not so common, it would be illegible for readers.
Response 1: I provided to replace abbreviations for things that were used infrequently with the respective extended definitions.
Point 2: The author misinterpreted the wording of the Japanese guideline, which recommends endoscopic treatment for HGD, not LGD.
Response 2: I’ve better explained the indications for Endoscopic resection of the Japanese guidelines at line 207.
Point 3: My opinion was on whether it was wrong to perform ESD or other aggressive treatments in the cases of LGD.
Response 3: This is also our opinion, therefore we did not dwell too much in the manuscript on this technique, resulting unnecessary in most cases of LGD.
Point 4: To be precise and detailed, it would be appropriate to compare CE-D between the LGD observation groups and the prior treatment groups over a long period. An argument should be made regarding this.
Response 4: I have added two additional columns in Table 2, specifying the complete eradication of dysplasia (CE-D) rate in patients treated with Radiofrequency Ablation group and in the surveillance group.
Point 5: As for Table 3, the problem is not that there are not enough papers, but that the table is not complete and may be difficult for readers to read. The comparison and target groups are not consistent, and the expressions were lacks politeness and care. Abbreviations and non-abbreviated expressions are mixed in the table. Could we really say that this table was complete? I know what “N circle” means. But is it common? I said that you had to fix it so that readers can easily understand this. Complication and pain do not need to be included in the table, they should be expressed in the text. Rather, please make a more comprehensive table comparing CE-D for ESD/EMR, RFA, CRYO, and APC.
Response 5: I’ve extended Table 3, including the studies reported in the manuscript that compare CE-D for EMR, RFA, Cryoablation techniques and Hybrid-APC, hoping to have made a more comprehensive and clear table.

Round 3
Reviewer 1 Report
This paper has been substantially revised after we pointed out some revisions.
If you don’t need any abbreviations, you have to delete abbreviations from the manuscript.
Again, the author misinterpreted the wording of the Japanese guideline, which recommends endoscopic treatment for HGD, not LGD. The LGD was interpreted as adenoma or well-differentiated adenocarcinoma with low-grade atypia (noninvasive). The guidelines weakly recommend that surveillance would have been conducted for pre-HGD conditions, such as LGD.
Table 3 became much easier to read. However, what is ”N circle”? Pages 343-345 must be revised accordingly.
Throughout the entire process, there have been discrepancies in the manuscript as revisions were made. I hope that you do not correct only parts but correct the whole sentences.
Author Response
Point 1: If you don’t need any abbreviations, you have to delete abbreviations from the manuscript.
Response 1: We have used abbreviations for the most used terms, such as Low-grade dysplasia, High-grade dysplasia, complete eradication of dysplasia, radiofrequency ablaion, to avoid too many repetitions in the manuscript and to make the text more fluent to read.
Point 2: Again, the author misinterpreted the wording of the Japanese guideline, which recommends endoscopic treatment for HGD, not LGD. The LGD was interpreted as adenoma or well-differentiated adenocarcinoma with low-grade atypia (noninvasive). The guidelines weakly recommend that surveillance would have been conducted for pre-HGD conditions, such as LGD.
Response 2: I’ve provided to change the reference to the Japanese guidelines, adding a clearer bibliographic reference (41) that better explains Japanese point of view about LGD.
Point 3: Table 3 became much easier to read. However, what is ”N circle”? Pages 343-345 must be revised accordingly.
Response 3: I replaced “n circle” with “number” in table 3 and I accordingly revised lines 343-345.
Point 4: Throughout the entire process, there have been discrepancies in the manuscript as revisions were made. I hope that you do not correct only parts but correct the whole sentences.
Response 4: I proceeded to modify the whole sentences subject to changes, hoping there are no more discrepancies between the various parts in the manuscript.
Round 4
Reviewer 1 Report
This article was well described the BE with LGD’s diagnosis and management. There is enough reference about the current evidence for BE with LGD.
The author has to delete the white letters about the patient in Figure 1 (b), though the editor will mention about that. Once you have presented abbreviations in earlier sentences, please use the abbreviations in later sentences.
Author Response
Point 1: The author has to delete the white letters about the patient in Figure 1 (b), though the editor will mention about that.
Response 1: I provided to delete the white letters in Figure 1 (b).
Point 2: Once you have presented abbreviations in earlier sentences, please use the abbreviations in later sentences.
Response 2: I replaced the extended definitions with the respective abbreviations after their first presentation in the manuscript.
